# Induced fit with replica exchange improves protein complex structure prediction

**Ameya Harmalkar** [ORCID], **Sai Pooja Mahajan** [ORCID], **Jeffrey J. Gray** [ORCID] *

Department of Chemical and Biomolecular Engineering, The Johns Hopkins University, Baltimore, Maryland, United States of America

* jgray@jhu.edu

**Data Availability Statement:** The source code for ReplicaDock 2.0 docking, with interface-tests, global and local docking examples and directed induced-fit, is available with Rosetta (www. rosettacommons.org). Scripts and tutorials are

## Abstract

Despite the progress in prediction of protein complexes over the last decade, recent blind protein complex structure prediction challenges revealed limited success rates (less than 20% models with DockQ score > 0.4) on targets that exhibit significant conformational change upon binding. To overcome limitations in capturing backbone motions, we developed a new, aggressive sampling method that incorporates temperature replica exchange Monte Carlo (T-REMC) and conformational sampling techniques within docking protocols in Rosetta. Our method, ReplicaDock 2.0, mimics induced-fit mechanism of protein binding to sample backbone motions across putative interface residues on-the-fly, thereby recapitulating binding-partner induced conformational changes. Furthermore, ReplicaDock 2.0 clocks in at 150-500 CPU hours per target (protein-size dependent); a runtime that is significantly faster than Molecular Dynamics based approaches. For a benchmark set of 88 proteins with moderate to high flexibility (unbound-to-bound iRMSD over 1.2 Å), ReplicaDock 2.0 successfully docks 61% of moderately flexible complexes and 35% of highly flexible complexes. Additionally, we demonstrate that by biasing backbone sampling particularly towards residues comprising flexible loops or hinge domains, highly flexible targets can be predicted to under 2 Å accuracy. This indicates that additional gains are possible when mobile protein segments are known.

## Author summary

Proteins bind each other in a highly specific and regulated manner, and these associated dynamics of binding are intimately linked to their function. Conventional techniques of structure determination such as cryo-EM, X-ray crystallography and NMR are time-consuming and arduous. Using a temperature-replica exchange Monte Carlo approach that mimics the kinetic mechanism of "induced fit" binding, we improved prediction of protein complex structures, particularly for targets that exhibit considerable conformational changes upon binding (Interface root mean square deviation (unbound-bound) > 1.2 Å. Capturing these binding-induced conformational changes in proteins can aid us in better understanding biological mechanisms and suggest intervention strategies for disease mechanisms.

available in the supplementary files as well as via demos in Rosetta.

**Funding:** This work was supported by National Institute of Health through grant R01-GM078221 (A.H) and R35-GM141881 (A.H, S.P.M and J.J.G). The funders had no role in study design, data collection and analysis, decision to publish, or preparation of the manuscript.

**Competing interests:** I have read the journal's policy and the authors of this manuscript have the following competing interests: J.J.G. is an unpaid board member of the Rosetta Commons. Under institutional participation agreements between the University of Washington, acting on behalf of the Rosetta Commons, Johns Hopkins University may be entitled to a portion of revenue received on licensing Rosetta software, which includes the methods described in this paper. JJG has a financial interest in Cyrus Biotechnology. Cyrus Biotechnology distributes the Rosetta software, which may include methods developed in this study. These arrangements have been reviewed and approved by the Johns Hopkins University in accordance with its conflict-of-interest policies.

This is a *PLOS Computational Biology* Methods paper.

## Introduction

Protein-protein interactions (PPIs) mediate most molecular processes in human health and disease, ranging from enzyme catalysis and inhibition to signaling and gene regulation. Predicting protein complex structures can aid in the systematic mapping of PPI networks in the cell, thereby revealing biological mechanisms and providing insights in protein structure-function relationships [1]. Experimental techniques can determine high-resolution protein structures, however, they can be expensive, laborious, and limited. Computational modeling of protein complexes, i.e., protein-protein docking, provides an alternative to elucidate structures and to identify putative interfaces.

The accuracy of most docking methods is hampered by binding-induced conformational rearrangements between protein partners [2]. The recent rounds of the community-wide blind docking experiment, Critical Assessment of PRediction of Interactions (CAPRI) [3, 4], showed that capturing large-scale conformational changes between protein partners (unbound to bound $C_\alpha$ root mean square deviation ($RMSD_{BU}$) > 1.2 Å) remains a longstanding challenge: Less than 20% of models submitted for these targets achieved a DockQ score [5] > 0.4 (see first figure in Harmalkar and Gray, 2021 [2]).

To improve docking performance, extensive sampling of the protein's backbone conformations is critical. Earlier studies have incorporated backbone motions either by docking a small ensemble (10–20) of backbone conformations of two proteins [6, 7] or by moving a restricted set of coordinates [8–10], but they obtained limited success, underscoring the need of better backbone sampling [11]. To push towards larger conformational changes, algorithms broadly emulate two kinetic binding models: (1) conformer selection (CS), and (2) induced-fit (IF) [12–14]. In CS, unbound protein monomers exist in an ensemble of diverse conformations, and the monomer conformations corresponding to the thermodynamically stable minima are selected upon binding [13]. This mechanism motivated our prior method, RosettaDock 4.0 Marze2018, a Monte-carlo (MC) minimization protocol that was efficient enough to use 100 pre-generated backbone structures of each unbound protein. RosettaDock 4.0 improved docking to highest reported success rates on flexible targets (49% of moderate, $RMSD_{BU}$ > 1.2 Å, and 31% of difficult targets, $RMSD_{BU}$ > 2.2 Å, successful predictions). However, since the performance of CS-based approaches depends on having native-like backbone conformations in the monomer ensembles, to capture binding-induced conformational changes, it is desirable to sample backbones in a partner-dependent fashion.

Induced-fit (IF) approaches incorporate partner-specific, localized conformational rearrangements. In IF, proteins 'induce' conformational changes upon molecular encounter [15, 16]. Since simulating backbone changes throughout the entire protein concomitantly with rigid-body perturbations is computationally expensive ($\mathcal{O}(2 \times 3^{N+1})$ as opposed to $\mathcal{O}(6)$ for $N$ atoms), IF docking approaches have typically been restricted to small backbone perturbations and side-chain movements [9, 17, 18]. Molecular dynamics (MD) simulations follow the IF-approach for all atoms, however, they are bound by time and length scales [19, 20]. Thus, expensive molecular dynamics (MD) simulations are accelerated with alternative sampling techniques such as steered MD [21], replica-exchange [22], or metadynamics [23] to refine rigid-body poses of docked proteins or dock small, rigid proteins [24, 25].

Replica exchange methods, in particular, have been employed for protein docking to perform an unprecedented sampling of putative protein complex structures [26] and association

pathways [21]. Temperature replica exchange methods modulate temperature across parallel replicas, with periodic exchanges between the high temperature replicas and the low temperature ones [22]. While temperature affects all atoms, Hamiltonian replica exchange methods update the energy function between the replicas and focus on a relevant degree of freedom of the system [25, 27, 28]. To date, however, none of these methods incorporate larger conformational rearrangements between protein partners upon docking. Moreover, most of the modeling examples have been limited to rigid-proteins with little flexibility ($RMSD_{BU} < 1.2$ Å).

Here, we couple the sampling prowess of replica exchange algorithms with the induced-fit binding mechanism to develop a new, aggressive, flexible backbone protein docking method. Our method, ReplicaDock 2.0, builds on Zhang *et al.*'s prior work on replica-exchange MC-based rigid-docking (ReplicaDock [22, 27]) and adds backbone motions along with a fast-scoring, low-resolution energy function to tackle moderate and highly flexible targets. We test our method on a diverse set of protein targets from the Dockground benchmark [29] that spans rigid, moderately flexible, and highly flexible targets. Despite the power of REMC, it is still unfeasible to explore all backbone conformational degrees of freedom, therefore we test the efficacy of choosing different flexible subsets. Finally, we examine whether biasing the sampling choices can generate sub-angstrom quality predictions.

## Materials and methods

### Energy function

**Low-Resolution energy function.** The low-resolution mode of the docking protocol utilizes score function built upon the existing six-dimensional, residue-pair transform dependent energy function, called the `motif_dock_score` [11]. To evaluate backbone sampling and penalize poor backbone conformations, we combine the `motif_dock_score` with energy terms that account for protein backbone dihedral conformations and torsion angles, such as `rama_prepro` (to evaluate backbone $\Phi$ and $\Psi$ angles), `omega` (to account for omega torsion corresponding to rotation about the C-N atoms), `p_aa_pp` (a knowledge-based score term that observes the propensity of an amino acid relative to the other amino acids) [30]. To account for inter- and intra-molecular clashes owing to on-the-fly backbone sampling, we also utilize a clash penalty based on atom-pair interactions (i.e. Van der Waals attractive and repulsive interactions). The updated score function, called Motif Updated Dock Score (MUDS), serves as the energy function for the low-resolution docking stage in ReplicaDock 2.0.

$$E_{\text{MUDS}} = E_{\text{motif−dock}} + E_{\text{LJrepulsive}} + E_{\text{LJattractive}} + E_{\text{backbone−statistics}}$$

**All-atom energy function.** To refine the docked outputs obtained from low-resolution docking, we use the standard all-atomistic energy function in Rosetta, called `ref_2015` an energy function based on physical, empirical, statistical and knowledge-based score terms [30].

### Generation of initial conformations

The docking challenge can be categorized dependent on two scopes, namely, (1) Global docking, where there is no *a priori* knowledge about protein binding, and (2) Local docking, where we have limited information about the binding regions. Global docking challenges are blind protein docking challenges involving prediction of the potential binding sites or orientations. After identifying potential binding regions, local docking aims to narrow down the scope to a localized region of protein to predict the conformations of complexes with better confidence.

ReplicaDock2.0 can be applied for both global docking and local docking. For a global docking search, the initial unbound conformations of the binding partners constitute the starting pose (structural model). To generate this starting pose, we randomize the initial orientation of the protein partners (unbound monomers prepacked with Rosetta FastRelax) with the Rosetta option `randomize1, randomize2 and spin` (details in the sample XML script). This orients a binding partner (say ligand), at a random orientation around the other binding partner (say receptor), resulting in a blind global docking set-up.

For local docking simulations, wherein the binding site or patch on the binding partners is known, we start by superimposing the unbound monomer structures over the bound structure. Then, we move the unbound monomers 15 Å away from each other with a 45˚) rotation to the ligand (smaller monomer) with respect to the receptor. This serves as the input structure to the ReplicaDock 2.0 protocol. For each trajectory, a Gaussian random 1 Å and 1˚ perturbation provides slightly different starting states. We have observed that higher temperature replicas often result in much broader exploration of the protein surface. The experimental bound structure is passed to the protocol as the native structure, and is employed as the reference for calculating the RMSDs. Further details about the protocols, command lines and scripts are reported in the S1 Text.

## ReplicaDock 2.0 protocol

To sample binding-induced conformational changes during docking, we employed a temperature Replica-exchange MC protocol with backbone conformational sampling in ReplicaDock 2.0. Backbone conformations are sampled with Rosetta Backrub [31]. Amongst the putative interface residues, two terminal residues for each contiguous fragment on the interface are chosen as pivots and backbone dihedral angles are sampled for the residues in between, thereby providing a restrictive IF-like motion.

We scale temperature across three replicas with inverse temperatures set to $1.5^{-1}$ kcal$^{-1}$.mol, $3^{-1}$ kcal$^{-1}$.mol and $5^{-1}$ kcal$^{-1}$.mol, respectively. Replica exchange swaps are attempted every 1,000 MC steps and candidate structures are stored after every successful swap. An exchange attempt is successful if the Metropolis criterion is obeyed as stated below:

$$P_i < \min \left\{ 1, \frac{\exp\left(\frac{-E_j}{k_B T_i} - \frac{-E_i}{k_B T_j}\right)}{\exp\left(\frac{-E_i}{k_B T_i} - \frac{-E_j}{k_B T_j}\right)} \right\}$$

Here, *i* and *j* are the replica-levels across which the swap is performed, *E* is the MUDS energy, $k_B$ is the Boltzmann's constant, T is temperature and $P_i$ is the probability set in Metropolis criterion that needs to be obeyed for acceptance (generally set to 0.5). Thus, ReplicaDock 2.0 simulations scale the temperatures to modulate the acceptance of backbone and docking moves, so motions that are penalized heavily at lower replicas can be accepted at higher replicas, thereby allowing more diversity in capturing backbone conformations as well as docking orientations. The generated candidate structures are further passed to the high-resolution stage for all-atom refinement. The all-atom, high-resolution refinement resolves any side-chain clashes and penalizes false-positive orientations that the low-resolution score function failed to penalize. This ensures that our output structures are at the lowest possible energetic state achievable for the attained conformations. For each local docking simulation, we initiate 8 trajectories, each trajectory spanning over 3 replicas, run for 2.5 x $10^5$ MC steps generating

$\sim$ 5,000 candidate structures. For global docking, we run $10^6$-$10^8$ MC steps, generating roughly 24,000 candidate structures.

## Benchmarking, evaluation and success metrics

Four interface residue selection tests were performed on 12 unbound targets from the Dock-Ground Benchmark Set [29] to optimize the flexibility scope over interface residues, number of trajectories and MC trials. Dockground Benchmark Set [29] classifies protein targets as rigid ($RMSD_{unbound-bound}$ < 1.2 Å), medium (1.2 Å ≤ $RMSD_{unbound-bound}$ ≤ 2.2 Å) and difficult targets ($RMSD_{unbound-bound}$ ≥ 2.2 Å), depending upon the conformational change between unbound and bound structures. ReplicaDock 2.0 docking runs were performed on the entire Dockground benchmark set of 44 medium and 34 difficult targets. We added 10 rigid targets for a final set with 88 targets. As defined in CAPRI [32], we calculated the interface RMSD (I-rms), ligand RMSD (L-rms), all-atom RMSD(RMSD), $C_\alpha$ RMSD and fraction of native-like contacts ($f_{nat}$) against the bound complex. Further, the results of the docking simulations were evaluated with the expected N5 metric. N5 denotes the number of near-native decoys in the five top-scoring structures. A structure is deemed as near-native if the $C_\alpha$ RMSD ≤ 5 Å for the low-resolution stage, and if CAPRI rank ≥ 1 for the high resolution stage [6]. First, we bootstrapped 1,000 structures, i.e. randomly selected 1,000 structures with replacement from the generated candidate structures. Then, by evaluating whether the five top-scoring structures were near-native, we determined the N5 value. This procedure was repeated 1,000 times for robustness to obtain the expected value ($\langle$N5$\rangle$). Successful docking for a target is defined as $\langle$N5$\rangle$ ≥ 3.

## Results

Protein-protein docking studies with T-REMC by Zhang *et al.* [22, 27] demonstrated significant improvement in sampling docking orientations, albeit with two important limitations. (1) No backbone degrees of freedom were sampled, restricting the search to rigid-body moves and thus precluding success on medium and highly-flexible docking targets. (2) The low-resolution energy function was inaccurate [22], so the improved sampling often led to incorrect complex structures. In this work, we address these limitations and improve protein-protein docking for previously intractable flexible targets.

## ReplicaDock 2.0 protocol selectively samples backbone degrees of freedom while docking

To address the backbone sampling limitation, we created ReplicaDock 2.0, an induced-fit (IF) inspired, T-REMC plus minimization algorithm that samples backbone conformations on-the-fly while docking. ReplicaDock 2.0 consists of two stages, low-resolution sampling and high-resolution refinement (Fig 1). To capture backbone degrees of freedom, the low-resolution stage performs replica-exchange and samples both backbone conformations and rigid-body orientations. For each docking pose sampled, backbone moves are sampled via *Rosetta Backrub* [31] over the interface residues. Our hypothesis is that by narrowing our search to the putative interface, the protocol will capture realistic conformational changes while maintaining feasible compute times. The low-resolution IF-based method samples the six rigid-body degrees of freedom along with the $3^N$ backbone degrees of freedom ($\phi$, $\psi$, $\omega$) for $N$ interface residues. By extending this sampling procedure over three replicas with inverse temperatures, $\beta$, of $1.5^{-1}$ kcal$^{-1}$ · mol, $3^{-1}$ kcal$^{-1}$ · mol and $5^{-1}$ kcal$^{-1}$ · mol, a range of backbone conformations sampled. We chose the number of replicas and replica-temperatures such that the energy distribution at any replica overlaps sufficiently with adjacent replicas, allowing efficient

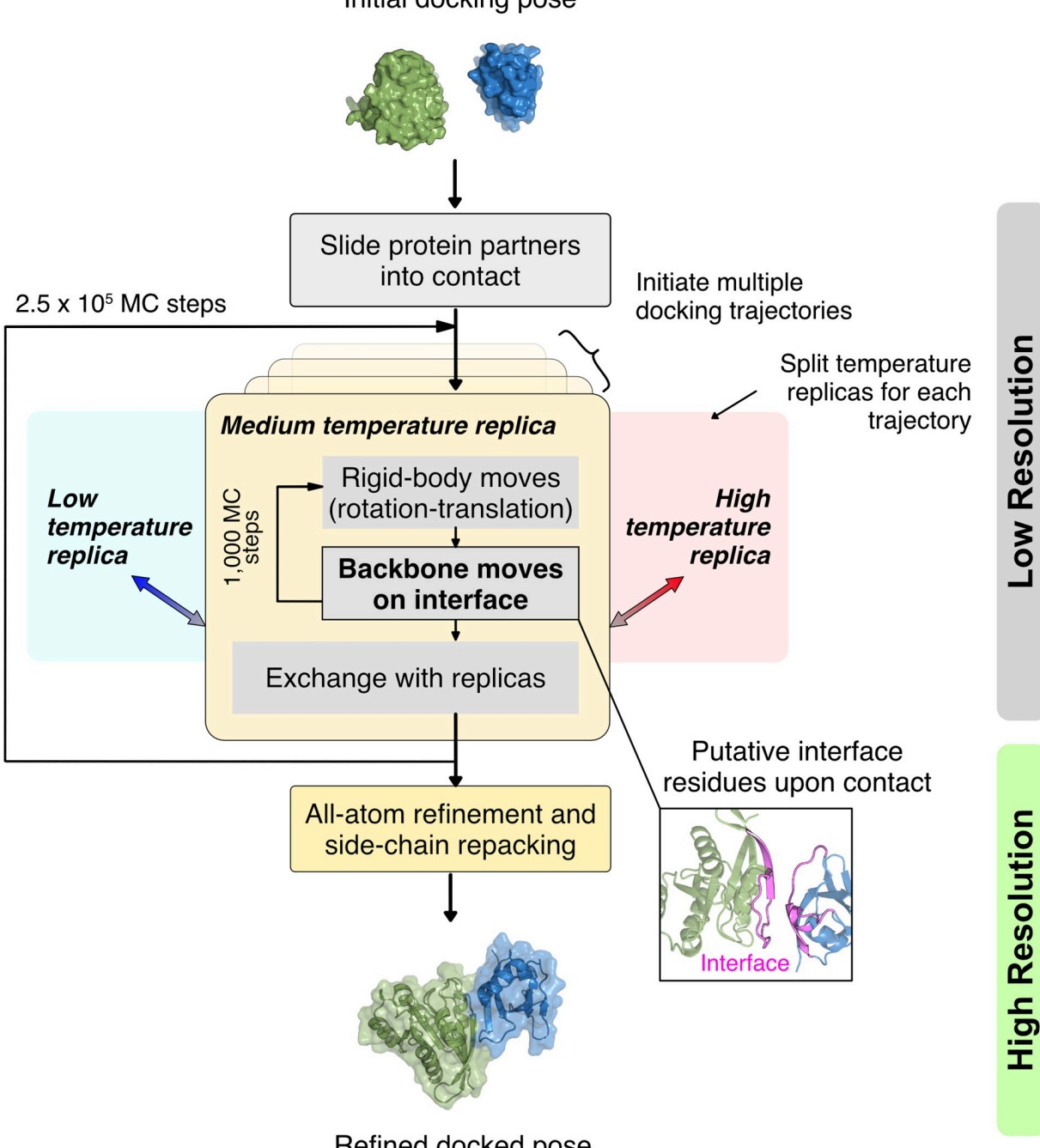

**Fig 1. Overview of the ReplicaDock2 protocol.** Starting from an initial docking pose i.e. a structural model with randomly oriented protein partners, the protocol perturbs the protein partners and slides them into contact. This creates an initial docking pose for the low-resolution stage. Here, the pose object is copied to three parallel replicas per trajectory, and each replica performs rigid body moves (rotation-translation) and backbone moves for each MC trial, followed by exchange between replicas after every $1,000^{th}$ trial. Each exchange obeys the Metropolis acceptance criterion and if accepted, the low resolution structure is output. Each trajectory completes $2.5 \times 10^5$ MC trial steps, and produces $\sim 5,000$ candidate structures. Lastly, all produced structures undergo an all-atom refinement comprising of side-chain packing, small rigid-body motions, and energy minimization to output final docked structural models.

exchanges (S1 Fig). After every 1, 000 MC trials of rigid-body and backbone motions, an MC-swap is attempted between neighboring replicas as per the Metropolis criterion [33] (Methods). Higher temperature replicas accept backbone moves that would be otherwise rejected at lower temperatures. To expand the diversity of sampled structures, up to 8 independent trajectories are initiated from the starting docking pose. After generation of candidate docking poses in the low-resolution stage, the high-resolution stage performs an all-atom refinement which employs finer rigid-body motions (random rigid-body perturbations in a Gaussian distribution of 0.1 Å to 3˚) with side-chain rearrangements followed by energy minimization in the torsional space. This stage does not explicitly move the backbone of the docked proteins but resolves any side-chain clashes and forms a compact, low-energy, high-resolution interface. After evaluating the refined structures' all-atom scores, the lowest scoring structure is the complex prediction.

## ReplicaDock 2.0 uses a residue-transform based scorefunction

As ReplicaDock 2.0 performs backbone sampling and generates docking poses during the low-resolution stage, it is crucial to have a score function that favors native-like interfaces. Thus, our next step in improving docking performance was to tackle the limitation of the inaccurate low-resolution centroid score function as observed by Zhang *et al.* [22]. In their recent CS-based approach, RosettaDock 4.0, Marze *et al.* [11] created the Motif Dock Score (MDS), a pre-tabulated score based on the residue-pair transforms approach [34] where energy of the interacting residues is defined by the 6-dimensional translation and rotation coordinates specifying their relative backbone locations. This simple scorefunction accurately estimated the well-tested all-atom score-function with a faster compute time [34]. MDS is restricted to inter-chain energies, which worked well for pre-generated monomer ensembles with fixed backbones in RosettaDock 4.0. For IF-based ReplicaDock 2.0, however, intra-chain energies must be included as the backbone moves, especially clashes. Therefore, we incorporated knowledge-based backbone torsion statistics terms and Van der Waals interaction terms [30] to create the Motif Updated Dock Score (MUDS). We optimized the relative weights of the MUDS energy terms based on the number of CAPRI acceptable quality structures in the top-scoring 10% of sampled structures (enrichment) (S6 Fig). With this updated scoring and sampling schemes, we tested the performance of our ReplicaDock 2.0 protocol for global and local docking tasks.

## Rigid global docking with ReplicaDock2.0 can identify local binding patches

Docking challenges can be categorized as either global (without any prior knowledge of binding interface) or local (using knowledge of putative binding patches). Conventionally, predictors search with a rigid protein backbone to identify putative binding interfaces (e.g., with ClusPro [35] or ZDOCK [36]), and then each binding interface is refined, often with backbone conformational change. This strategy breaks down docking hierarchically. Global docking has been performed with a T-REMC approach (ReplicaDock [22]), but low-scoring structures were often far from the experimental structure owing to the inaccurate centroid score function. With our updated score function MUDS, we hypothesized that its discriminative power would enable a rigid-body global docking simulation to better identify native-like interfaces. To test this hypothesis, we ran ReplicaDock 2.0 without backbone conformational sampling (only rigid-body rotational and translational moves) on 10 protein targets starting from random orientations of the protein partners.

To illustrate the rigid global docking performance, Fig 2A plots the low-resolution score (MUDS) versus the RMSD from the native structure for all generated candidate structures for

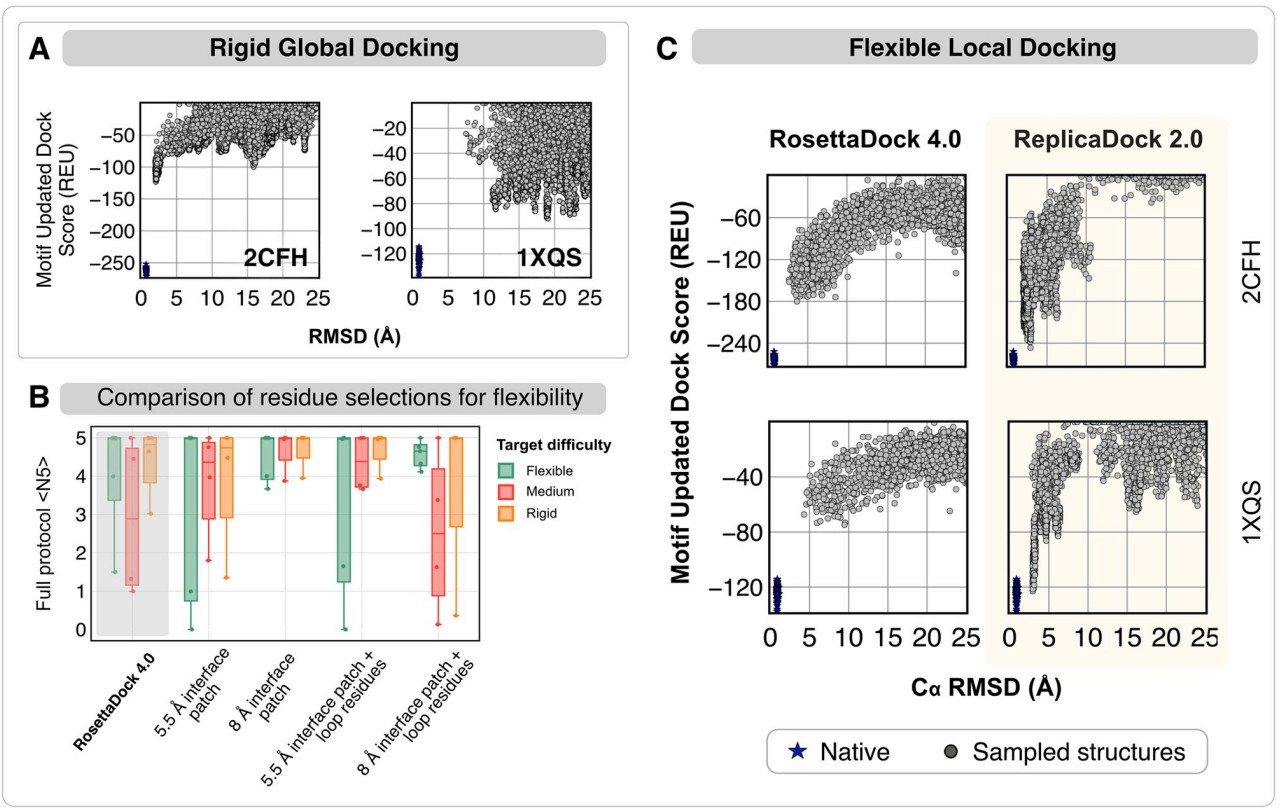

**Fig 2. T-REMC improves low-resolution performance in global rigid-body and local flexible docking for two representative protein targets.** *(A) Global rigid-body docking performance* for protein targets 2CFH (trafficking protein particle complex subunits) [37] and 1XQS (HspB1 core domain complexed with Hsp70 ATPase domain) [38]. Plots show the Motif Updated Dock Score (REU) vs all-atom Cα rmsd (Å). Blue points denote the refined native structures. *(B) Comparison of different residue selections for performing backbone moves.* Performance of ReplicaDock 2.0 with four conditions: (1) 5.5 Å interface patch, (2) 8 Å interface patch (3) 5.5 Å interface patch + loops, (4) 8 Å interface patch + loops. The metric is ⟨N5⟩, the average number of near-native models in the five top-scoring structures. For reference, RosettaDock 4.0 performance is highlighted in gray. *(C) Local flexible backbone docking performance.* Motif Updated Dock Score (REU) vs Cα rmsd (Å) for two targets, 2CFH [37] and 1XQS [38]. Panels show ~5,000 decoys generated by RosettaDock 4.0 (left) and ReplicaDock 2.0 (right, this work).

two representative, medium-flexibility protein targets (2CFH, trafficking protein particle complex subunits, 1.55 Å $RMSD_{BU}$ [37] and 1XQS, HspB1 core domain complexed with Hsp70 ATPase domain, 1.77 Å $RMSD_{BU}$) [38]. As a reference, we relaxed the experimental bound structure with relatively small rigid-body moves (rotations and translations of 0.5° and 0.1 Å, respectively) to generate near-native structures (blue in Fig 2A).

ReplicaDock 2.0 generates low-scoring near-native orientations (under 5 Å RMSD) for 2CFH, however, for 1XQS, sampling is limited to RMSD values above 6 Å, with the lowest scoring structures about 20 Å away from the experimental structure. On 10 protein targets (S2 Fig), ReplicaDock 2.0 produced models within 5 Å of the native-bound structure for 8 of 10 targets. For comparison, ClusPro [39] successfully predicts 6 of 10 targets. Thus, ReplicaDock 2.0 can perform exhaustive global sampling on the protein energy landscape with better near-native discrimination. One limitation is that global docking with ReplicaDock 2.0 requires 600–800 CPU hours, compared with 35 CPU hours (as reported by Varela *et al.* [40]) for ClusPro [35]. Rigid-backbone global docking results from either ClusPro or ReplicaDock 2.0 can serve as the input to a local, flexible-backbone docking search. (AlphaFold [41] or AlphaFold-Multimer [42] could also be used to generate starting structures [43, 44] for refinement, if the

multiple sequence alignments are sufficient for the target. We discuss some comparisons for past CASP14-CAPRI targets [45] in S3 and S7–S9 Figs.

## Flexible local docking with ReplicaDock2.0 samples deeper energy funnels

When given a putative, broadly-defined binding patch, local docking approaches strive to obtain the biological complex structure by capturing conformational changes in protein partners. ReplicaDock 2.0 explores conformational changes by restrictively sampling backbone moves at putative interfaces. To evaluate the extent of flexibility that can be incorporated while docking for optimum performance, we tested ReplicaDock 2.0 protocol (low resolution sampling with high resolution refinement) with different selections of residues for backbone sampling (S4 Fig). First, we performed backbone moves conservatively over only the set of residues with atoms lying within 5.5 Å of the binding partner (Set 1: 5.5 Å interface patch). Then, we expanded the selection to residues with atoms lying within 8 Å of the binding partner (Set 2: 8 Å interface patch). As loops are the most flexible secondary structural element in a protein structure, we incorporated residues belonging to all the loop regions from the unbound protein monomers, and added them to prior residue sets to obtain Set 3 (5.5 Å interface patch + loop residues) and Set 4 (8 Å interface patch + loops residues) respectively. For local docking on 12 test targets, we generated $\sim$5,000 structures and sub-sampled sets of 1,000 structures to calculate the expected number of near-native structures (defined as CAPRI acceptable quality or better) in the 5 top-scoring structures ($\langle N5 \rangle$). $\langle N5 \rangle$ evaluates the ability of a protocol to sample near-native conformations and discriminate them from false-positive structures (see Methods). Higher $\langle N5 \rangle$ indicates that in blind predictions, top-scoring structures are more likely to be correct. Fig 2B compares traditional CS-based RosettaDock 4.0 performance with IF-based ReplicaDock 2.0 using each of the four flexibility scopes. Extending the backbone moves to 8 Å interface patch increased $\langle N5 \rangle$ across all targets, and offered enough flexibility to capture the binding-induced conformational changes. Incorporating loops reduced performance for medium-flexible and rigid targets (average performance for medium-flexible targets dropped from $\langle N5 \rangle = 5$ in Case 2 to $\langle N5 \rangle = 2.5$ in Case 4), possibly due to over-sampling of backbone moves in relatively rigid regions of the protein structure. Adding flexibility to all loops, the scorefunction misdirects sampling in non-native, spurious minimas, resulting in alternate binding modes with large buried surface area or distorted protein tertiary structures (as shown by the false positive minimas in S5 Fig). To capture realistic backbone conformations, we therefore restrict backbone moves to an 8 Å local interface. Unfortunately, this selection precludes longer range, off-site conformational changes.

With the 8 Å selection chosen as the mobile residue set, we next evaluated the local docking performance of ReplicaDock 2.0 against RosettaDock 4.0. This also served as a head-to-head comparison between two kinetic mechanisms of binding i.e. IF versus CS. As an example, Fig 2C shows the generated candidate structures for two representative protein targets 2CFH and 1XQS with the two docking methods. The low-resolution score (MUDS) versus $C_\alpha$ RMSD plots for the targets 2CFH and 1XQS show that ReplicaDock 2.0 samples structures that score lower than RosettaDock 4.0. Further, in contrast with RosettaDock 4.0 funnels, ReplicaDock 2.0 produces deeper funnels, suggesting that as induced-fit enables the protocol to capture better backbone conformations, replica exchange improves the docking orientations of the encounter complexes generated, thereby allowing us to reach lower, native-like energies (bound-derived funnel in blue).

## Induced-fit recapitulates native contacts but fails to push backbone sampling towards bound conformations

With low-resolution sampling, ReplicaDock 2.0 explores larger conformational space in a rapid fashion and avoids entrapment in local minima. However, the structures generated are

limited with their accuracy and often require all-atom refinement to penalize side-chain clashes or spurious interfaces and yield realistic structures. The all-atom refinement can further lead to smaller motions and side-chain rearrangements that can result in compact binding between protein partners. Hence, we refined the candidate structures generated in low-resolution stage with the Rosetta all-atom `ref2015` energy function [30]. S10–S12 Figs represent the low-resolution candidate structures colored with their final high-resolution CAPRI quality. In multiple cases (e.g., medium targets 1MQS and 2HRK), the high-resolution stage penalizes poor, false-positive structures and refines near-native structures to improve their quality, showing that best results are achieved by combining the MUDS T-REMC stage with all-atom refinement.

Fig 3A and 3D highlight the high-resolution performance for the same two protein targets (2CFH and 1XQS) by comparing the interface energies (equivalent to thermodynamic binding energies) versus the interface-RMSD. For both protein targets, ReplicaDock 2.0 retains the better-scoring structures from the low-resolution stage (Fig 2C). Relative to RosettaDock 4.0, ReplicaDock 2.0 structures have better all-atom scores and an improved CAPRI quality as evident by the greater number of medium-quality decoys. Despite this improvement, there remains a gap in interface-RMSD between the lowest scoring docked structures and the refined native structures (blue in Fig 3A and 3D). To determine how induced-fit affects the backbones, we calculated the monomer component backbone RMSDs from the bound backbone conformations (Fig 3B and 3E). Although ReplicaDock 2.0 generates a much more diverse set of backbone conformations than RosettaDock 4.0, the best RMSDs attained by both the methods are comparable. Note that RosettaDock 4.0 uses pre-generated ensembles resulting in all candidate docking structures being limited in the backbone conformation space (all RMSDs within a rectangular region), whereas ReplicaDock 2.0 generates more diversity. These docking metrics for the entire benchmark set are illustrated in S13–S15 Figs. Further, we calculated the native-like interactions made by the interface residues with the fraction of native residue-residue contacts, $f_{nat}$ (Fig 3C and 3F). With the induced-fit strategy, ReplicaDock 2.0 increases the $f_{nat}$ over RosettaDock 4.0 by $\sim 0.2$. By sampling protein conformations in the vicinity of its binding partner, ReplicaDock2.0 is able to orient more interface residues to a native-like state, thereby recapitulating a larger fraction of bound contacts (S13–S15 Figs, on left).

## Benchmark evaluation demonstrates improved performance over conformer-selection methods

To evaluate the accuracy of local docking with ReplicaDock 2.0, we benchmarked our model on 88 protein targets from Docking Benchmark DB5.5 [29], constituting 10 rigid targets along with all 44 moderately-flexible (medium) and 34 highly flexible (difficult) targets. For each target, we generated $\sim 5,000$ candidate structures with ReplicaDock 2.0 and, for comparison, RosettaDock 4.0. The ensemble generation and pre-packing for the RosettaDock 4.0 protocol was performed as described in Marze *et al.* [11]. For ReplicaDock 2.0, we docked protein targets as summarized in **Methods**.

To compare the performance, we measured $\langle N5 \rangle$ after the low-resolution and high-resolution stage for the full benchmark set of 88 targets. We define a structure as near-native if the $C_\alpha$ RMSD $\leq 5$ Å for the low-resolution stage, and if the CAPRI rank is acceptable or better for the high resolution stage. Fig 4A shows the $\langle N5 \rangle$ scores of the benchmark targets for the two protocols. The dashed lines demarcate the region of no improvement i.e., the two protocols differ by less than one point in their $\langle N5 \rangle$ scores. For targets above the dashed line (upper diagonal region), ReplicaDock 2.0 performs better, while for those below the dashed line (lower

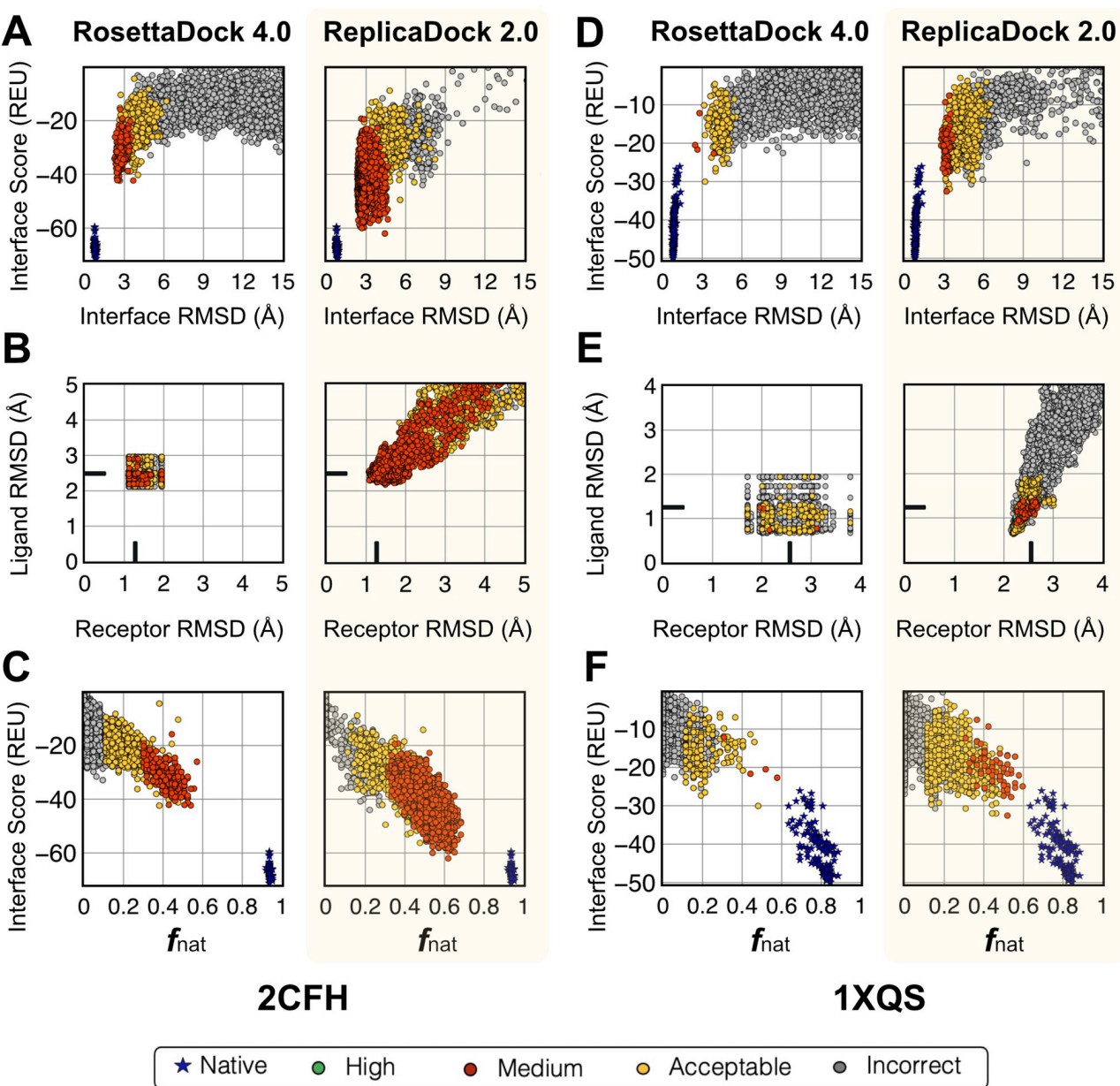

**Fig 3. Improvement in docking performance after full protocol for two representative targets.** (A,D) Interface score (REU) vs I-rmsd (Å), (B,E) Ligand-RMSD(Å) versus Receptor-RMSD(Å), and (C,F) Interface score (REU) vs fraction of native-like contacts post all-atom refinement for RosettaDock 4.0 [11] and ReplicaDock 2.0(this work) for two targets 2CFH and 1XQS. Relative to RosettaDock 4.0, ReplicaDock 2.0 samples decoys that score better, are closer to the native, have higher native-like contacts($f_{nat}$) and better CAPRI quality. However, backbone RMSDs (B,E) have not moved closer to the native but rather diverged away from it.

diagonal region), RosettaDock 4.0 performs better. In the low-resolution stage, ReplicaDock 2.0 outperforms RosettaDock 4.0 with nearly a third of the targets having better $\langle N5 \rangle$ (27 out of 88). After the high-resolution stage, ReplicaDock 2.0 outperforms RosettaDock 4.0 on 24 targets.

To better illustrate the trend, we plotted the probability density of $\langle N5 \rangle$ across all targets (Fig 4B). With the probability density curves, since the criterion to determine a near-native structure is different for high-resolution and low-resolution stages, the area under curve

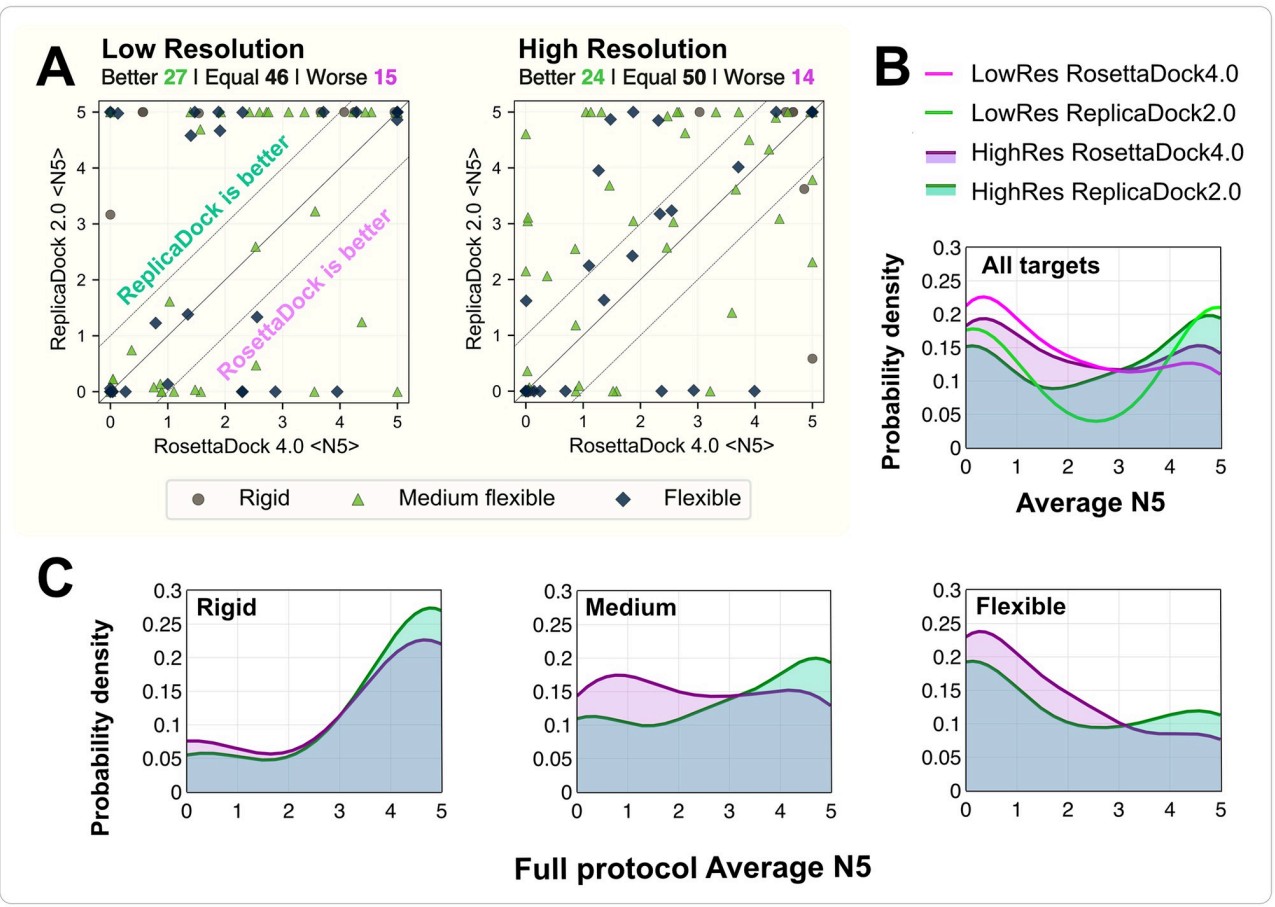

**Fig 4. Comparison of performance metrics between RosettaDock 4.0 and ReplicaDock 2.0 for individual complexes in a benchmark set of 88 docking targets.** (A) Comparison of ⟨N5⟩ values after low-resolution and high-resolution stages (full protocol), respectively. Dashed lines highlight the region in which the two protocols differ significantly, i.e. by more than one point in their ⟨N5⟩ values. Different symbols correspond to each target's difficulty category (circle: rigid; triangle: medium; diamond: flexible). Points above the solid line represent better performance in ReplicaDock 2.0, while points below the line represent better performance in RosettaDock 4.0. After the full protocol, 24 targets are modeled significantly better and 14 complexes are modeled significantly worse. (B) Probability density curves versus ⟨N5⟩ for all targets for ReplicaDock 2.0 (green) and RosettaDock 4.0 (purple). Low-resolution performance is indicated by lines (bright pink and bright green), and high-resolution performance is denoted by shaded area (purple and green). (C) Probability density curves versus full-protocol average N5 for rigid, medium and flexible targets respectively.

(AUC) differs. However, it can capture some overarching trends in performance: ReplicaDock 2.0 shifts the curve towards higher ⟨N5⟩, particularly for moderately-flexible targets ([Fig 4C]). For 37 out of 44 moderately-flexible targets, ReplicaDock 2.0 performance is either equivalent or better than RosettaDock 4.0. However, for highly flexible targets, the improvement is modest; docking proteins with higher conformational changes ($RMSD_{BU} > 2.2$ Å) is still a challenge. On an absolute basis with ⟨N5⟩ ≥ 3 as a success criteria, ReplicaDock 2.0 correctly predicts near-native docked structures in 80% of rigid, 61% of medium-flexible and 35% of highly-flexible docking targets.

Finally, we also compared the run time of ReplicaDock 2.0 with RosettaDock 4.0 for local docking across the benchmark targets. For all benchmark targets, we could generate the ReplicaDock 2.0 trajectories on our current hardware (24 processors) in a compute time of 8–72 CPU-hrs. The scaling of the ReplicaDock 2.0 and RosettaDock 4.0 protocols with the number of residues in the complex, $N_{res}$, is illustrated in [S17 Fig].

## Sampling of known mobile residues captures near-bound conformations of highly flexible protein targets

While ReplicaDock 2.0 generates better quality structures, it fails to reach sub-angstrom interface accuracy for many flexible targets, as shown in Fig 3A and 3C for 2CFH and 1XQS. Upon inspection of the bound and unbound structures of medium and difficult targets, we observed that the backbone conformational changes were diverse, ranging from motion of loops and changes in the secondary structure to hinge-like motion between intra-protein domains. The residue sets for backbone sampling in ReplicaDock 2.0 were not broad enough to capture these conformational changes. To push towards these larger backbone motions, we wondered whether ReplicaDock 2.0 might attain native-like backbones if it used the information of the residue set that results in the conformational change. To test this claim, we identified the mobile residues on the unbound protein partners of Ras:RALGDS domain complex (1LFD, 1.79 Å $RMSD_{BU}$ [46]) that showed more than 0.5 Å RMSD when superimposed over the bound structure. Next, instead of automating the selection of interface residues on-the-fly in the baseline protocol, we fed the ReplicaDock 2.0 protocol the identity of these mobile regions. In this version, we restricted the replica-exchange backbone sampling strategy towards pre-selected mobile residues, thereby implementing a *directed* induced-fit mechanism for protein docking.

To investigate whether directed induced-fit improves the docking performance, we evaluated the interface scores, native-like contacts and near-bound backbone conformations. Fig 5 compares the directed IF approach (*bottom*) with the vanilla version (*top*), which performs unbiased backbone sampling over putative interface residues. The results from Fig 5A and 5E

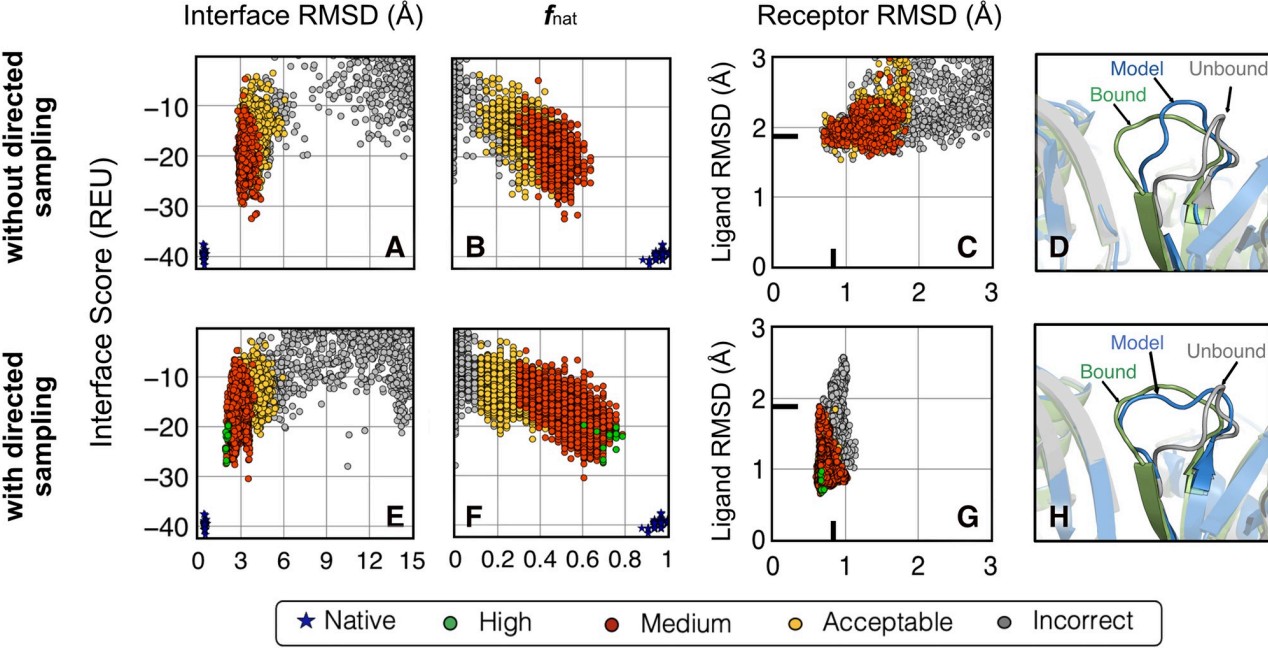

**Fig 5. Directed induced-fit improves flexible protein docking performance.** (top) (a,b,c) ReplicaDock 2.0 without directed backbone sampling of putative interfaces i.e. unbiased moves, finds medium-quality structures (colors: green = high quality, red = moderate quality, yellow = acceptable quality, gray = incorrect) (bottom) (e,f,g) ReplicaDock 2.0 with directed backbone sampling of mobile residues improves protein docking and obtains high-quality structures. (d,h) Comparing with the Ras' unbound structure (grey) superimposed over the bound (green), the docked structure loop (blue) has moved closer to the bound state (green) for the two cases respectively. With directed sampling, it is able to capture the backbone structure to sub-angstrom accuracy.

suggest that with directed IF, the protocol is now able to generate sub-angstrom structures with high-quality CAPRI ranks. In addition, Fig 5B and 5F show that it also increases the fraction of native-like contacts at the interface from an $f_{nat}$ score of roughly 0.6 to 0.8. The most significant difference is illustrated by the backbone RMSDs of the ligand and receptor chains relative to the bound structure. With directed sampling, the backbone RMSD values do not go higher than ∼0.5 Å away from the bound, starting from the unbound (Fig 5G), whereas the unbiased case samples extensive conformation space away from both bound and unbound (Fig 5C).

Finally, to give a structural perspective, Fig 5D and 5H show a cartoon-model representation of the unbound and model structure superimposed over the bound structure. With directed induced-fit, the flexible loop retraces an orientation similar to the bound structure. The protocol similarly identified high or medium-quality structures for 15 flexible test targets (detailed metrics in S16 Fig). Thus, if the flexible residue set could be better identified from the unbound structure, ReplicaDock 2.0 could improve docking further for flexible targets.

## Conclusion

In this work, we built on advances in T-REMC methods to develop a docking protocol that mimics induced-fit motion and effectively predicts protein complex structures upon binding. We determined that our IF-based docking protocol, ReplicaDock 2.0, generates more native-like structures than the state-of-the-art CS-based docking method, RosettaDock 4.0, on a benchmark set of moderate and difficult targets. With this work, we made two key advances. First, the updated scoring function (MUDS) recognizes native-like interfaces better and penalizes candidate structures with intra/inter-residue clashes, less frequent conformations or low thermodynamic stability. Second, ReplicaDock 2.0 augments the conventional REMC approach with backbone sampling. The protocol explores the ability of an induced-fit approach to manipulate backbone flexibility on docking by flexing interface residues with Rosetta Backrub [31]. Our studies demonstrate that instead of pre-configuring backbones for protein-docking (i.e., conformer selection), partner-dependent conformational changes (i.e., induced-fit) can result in better molecular recognition.

ReplicaDock 2.0 can be employed for both global and local docking simulations. We demonstrated that the global docking performance of ReplicaDock 2.0 is often better or at par with one leading global docking method (ClusPro), albeit requiring considerably more compute time. With local docking, ReplicaDock 2.0 consistently produced higher success rates using a stringent success criteria. We expand the table created by Marze *et al.*, to compare our results with six other leading docking methods: HADDOCK [47], ClusPro [35], iATTRACT [9], ZDOCK [36, 48], RosettaDock 3.2 [6] and RosettaDock 4.0 [11]. S1 Table compares these docking methods, their results and success metrics as well as the size of their benchmark set. Analogous to recent blind prediction challenges, the predictor methods perform with acceptable accuracy for rigid targets, however, the accuracy exceedingly drops as flexibility increases. ReplicaDock 2.0 improves the accuracy for docking moderate flexible targets to 61%, a significant increase over RosettaDock 4.0 (49%). On difficult targets, the improvement is still limited at 35%, a meager increase over RosettaDock 4.0 (31%). To the best of our knowledge, we present the first instance of a protein docking algorithm attaining ∼60% success rate on moderately flexible targets (1.2 Å < $RMSD_{BU}$ < 2.2 Å).

As ReplicaDock 2.0 is restricted to backbone sampling at putative interfaces, it fails to accommodate off-site conformational changes, for e.g., co-evolutionary residues triggering off-site domain motions or maintaining a fold in the tertiary protein structure. By directing backbone torsional sampling over known mobile residues, we observed that ReplicaDock 2.0 protocol

substantially improves the quality and accuracy of docking predictions. Thus, for blind targets, if we could identify potentially flexible residues from homologous structures or from AlphaFold's confidence metrics, such as predicted LDDT C$\alpha$ score (pLDDT) or predicted TM Score (pTM) [41, 42], ReplicaDock 2.0 could be guided to allow targeted flexibility. We anticipate that by improving our ability to predict intrinsic flexibility of residues, T-REMC docking with Replica-Dock 2.0 has potential to make even larger strides in flexible-backbone protein docking.

Despite the improvement in docking performance, ReplicaDock 2.0 brings limited computational speed-up. Currently, ReplicaDock 2.0 generates local docking structures in approximately 30–60% less time than RosettaDock 4.0. Yet, the high compute time is a caveat of the protocol, particularly for larger complexes, or complexes with higher interacting residues. As opposed to embarrassingly parallel approaches that can utilize higher compute power for a similar time-frame, ReplicaDock 2.0 requires 8–72 hrs on 24 processors for a docking simulation. Although this is computational efficient over conformer-selection methods, such as RosettaDock 4.0 or MD simulations, by increasing sampling trajectories and utilizing multiple processors, we could improve run times without compromising on backbone sampling.

Binding-induced conformational changes, and backbone flexibility at large, has long confounded protein-docking algorithms [2, 49]. Protein-protein docking with backbone flexibility via induced-fit for moderate to large-scale motions has not yet been reported. ReplicaDock 2.0 mimics induced fit by moving the backbone in conjunction with docking—for the first time—to consistently reach motions beyond 1 Å in the backbone during protein binding. By improving our understanding of protein interactions and the molecular recognition process, we could determine structures that are yet to be experimentally validated e.g., SERCA-PLB transmembrane complex critical for cardiac function [50], and explore potential association pathways, such as the translocation of protein antibiotics (e.g., colicins) through cellular nutrient transporters [2]. Insights into protein docking and binding interfaces have enabled successful computational designs such as symmetrical oligomers for self-assembling nanocages [51, 52] and orthogonal designs of cytokine-receptor complexes [53]. Capturing larger conformational changes will eventually impact our ability to design proteins with complex functions. Looking ahead, we anticipate that capturing the dynamic behaviour of proteins in docking will guide molecular engineering and de novo interface modelling to develop functional protein interfaces for biology, medicine and engineering.

## Supporting information

All references for the supporting information are listed in S1 Text.
**S1 Text. Supporting results and supporting methods.**
(PDF)

**S1 Table. Comparison of leading docking methods with ReplicaDock 2.0 (derived from Marze _et al._ [15]).** (1) Nearest-native structures from rigid-body docking selected for refinement. (2) Half successes awarded for targets with multiple binding sites evaluated, where at least one but not all binding sites are captured. (3) 2.5 Å cutoff for near-native structures. (4) Cases where bootstrapping gives ≥50% chance of N5≥3 are considered successfully docked. (5) For CAPRI sets, medium and difficult targets are combined, comprising all targets without at least one high-quality prediction by any predictor. (6) Lensink _et al._ [14] (7) Hwang _et al._ [13] (8) Vreven _et al._ [12]. The ReplicaDock 2.0 and RosettaDock 4.0 test sets differ slightly because we omitted some easy targets and we added flexible targets that had been too large for the prior ensemble methods.
(PDF)

**S2 Table. Performance of RosettaDock 4.0 vs. ReplicaDock 2.0 across an 88-target benchmark set.** 5,000 decoys were generated by each protocol for each target. Bootstrapped N5 values (plus standard deviations), both after the low-resolution phase and after the full protocol, are listed for each target. Success is defined as $\langle N5\rangle \geq 3$ for the N5 metrics.
(PDF)

**S1 Fig. Energy distribution of conformations sampled with RosettaDock 4.0 and ReplicaDock 2.0 (at respective inverse temperatures) for protein target 2CFH.**
(PDF)

**S2 Fig. Global docking performance.** Interface score (REU) vs I-rmsd(Å) for each of the 10 benchmark targets, arranged by target difficulty. ReplicaDock 2.0 decoys colored in gray and ClusPro models, relaxed with Rosetta and scored with MUDS, highlighted in red.
(PDF)

**S3 Fig. ReplicaDock2 and AlphaFold performances on blind CAPRI target T164.** (A) ReplicaDock2.0 prediction (green-blue) superimposed over the bound structure (in gray-tan). Comparison of interface regions between for bound (gray-tan) with ReplicaDock2.0 model (green-blue) and AlphaFold (pale green-pale blue) highlighted below. AlphaFold prediction (pale green-pale blue) superimposed over the bound structure. Binding orientation correlates with the wildtype, but prediction derives mostly from the unbound templates. AlphaFold-multimer was used via available jupyter notebook, but the models generated with AlphaFold-multimer were almost identical (less than 0.5 Å all-atom RMSD difference) to the models produced with AlphaFold. (B) Top-5 predictions for the predictor groups colored by the prediction quality. AlphaFold and ReplicaDock2 capture better quality decoys than predictions. RoseTTAFold predictions are inaccurate. (C) Interface-RMSD (Å) for the top-5 predictions by the predictor groups. RoseTTAFold models could not capture models within 8 Å and are excluded from this plot.
(PDF)

**S4 Fig. Interface residue selections (*in magenta*) highlighted over a protein target (Receptor, *in green* and ligand, *in blue*).** The four residues selections are as follows: (1) 5.5 Å interface patch, (2) 8 Å interface patch (3) 5.5 Å interface patch + loops, (4) 8 Å interface patch + loops. Note that, we also performed a test set by including all the residues of the protein for backbone sampling, however, with T-REMC, such simulations resulted in distortion of the protein quaternary structure (i.e. resulted in protein unfolding). Therefore, we chose to exclude that test.
(PDF)

**S5 Fig. MUDS versus C$\alpha$-RMSD(Å) plots for RosettaDock 4.0 and ReplicaDock 2.0 for two sets of residue selections in the low-resolution stage.** Mobile residues sets are as follows: (left) 8 Å interface patch, and (right) 8 Å interface patch + loops. Candidate structures are colored by the CAPRI quality post all-atom refinement (colors: green = high quality, red = moderate quality, yellow = acceptable quality, gray = incorrect).
(PDF)

**S6 Fig.** A. **Near-native enrichment** for two different weights based on MUDS. Each score-term is represented with the boxplot illustrating the enrichment of CAPRI-quality accceptable models within the lowest-scoring 1,000 models out of a set of 10,000 (10%) for each of the 11 protein-protein complexes. The motif-dock score corresponds to the score function defined in *Marze et al.* [15]. Motif Updated Dock Score(MUDS) *in blue* (as defined prior) comprises of Van der Waals score terms(attractive and repulsive clash terms) *in pale orange* and Backbone

statistical terms(`rama`, `p_aa_pp` and `omega`) *in pale green* along with the motif-dock score *in purple*. (left) Represents reduced weights of 0.1X to the Van der Waals and the backbone score terms, where X is the standard weight as defined by `beta_nov16`. (right) Represents a weight of 1.0X to the score-terms where X is the standard weight defined by the `beta_nov16` score-function. Sampling performance is improved almost three-fold for all 11 targets by improved weights. B. Distribution of protein targets to compare scoring performance.
(PDF)

**S7 Fig. AlphaFold complex modeling for CASP14-CAPRI target T164, comprising of the SMCHD1 (human) residues 1616–1899 Structural maintenance of chromosomes flexible hinge domain-containing protein 1.** (A) Native structure (PDB ID: 6N64) in grey, superimposed by the available complex template (PDB ID: 1GXK) in light-blue, with an interface RMS of 3.84 Å. (B) AlphaFold model (green-blue) superimposed over the complex template (in light-blue). AlphaFold models the protein complex closer to the complex template, with considerable conformational differences amounting to an interface RMSD of 2.16 Å. (C) AlphaFold model superimposed over the native shows an interface RMSD of 3.7 Å. Although marked as an easy target due to the availability of the template, this target comprises flexible loops and helical rearrangements.
(PDF)

**S8 Fig. ReplicaDock 2.0 performance on target T164 with and without AlphaFold template.** (A) Interface Score (REU) vs Interface-RMSD (Å) plots show that ReplicaDock captures acceptable decoys (and a few medium-quality decoys) starting from the complex template, however, if flexible docking is initiated with AlphaFold structures, we improve the quality of our predictions by 0.6–0.8 Å and capture more medium-quality decoys with better interface scores. (B) The models from ReplicaDock with (green-blue) and without(palegreen-paleblue) AlphaFold template are superimposed over the native(grey-tan).
(PDF)

**S9 Fig. ReplicaDock 2.0 performance on target T165 (a monoclonal antibody 93k bound to varicella-zoster virus glycoprotein gB).** AlphaFold predictions for complex predictions involved broken tertiary structures and are not reported. ReplicaDock found acceptable quality targets, however, the H3 loop was not adequately sampled owing to the relatively lower Interface-RMSD.
(PDF)

**S10 Fig. Score versus C$\alpha$-RMSD(Å) plots in the low-resolution stage for motif updated dock score with RosettaDock 4.0 and ReplicaDock 2.0 for rigid docking targets.**
(PDF)

**S11 Fig. Score versus C-RMSD(Å) plots in the low-resolution stage for motif updated dock score with RosettaDock 4.0 and ReplicaDock 2.0 for medium docking targets.**
(PDF)

**S12 Fig. Score versus C-RMSD(Å) plots in the low-resolution stage for motif updated dock score with RosettaDock 4.0 and ReplicaDock 2.0 for difficult docking targets.**
(PDF)

**S13 Fig. Interface Score versus Interface-RMSD(Å) plots and Interface Score versus $f_{nat}$ plots after the complete protocol for RosettaDock 4.0 and ReplicaDock 2.0 for rigid docking targets.**
(PDF)

**S14 Fig. Interface Score versus Interface-RMSD(Å) plots and Interface Score versus $f_{nat}$ plots after the complete protocol for RosettaDock 4.0 and ReplicaDock 2.0 for medium docking targets complexes.**
(PDF)

**S15 Fig. Interface Score versus Interface-RMSD(Å) plots and Interface Score versus $f_{nat}$ plots after the complete protocol for RosettaDock 4.0 and ReplicaDock 2.0 for difficult docking targets.**
(PDF)

**S16 Fig. Interface Score versus Interface-RMSD(Å) plots and Interface Score versus $f_{nat}$ plots after the complete protocol with RosettaDock 4.0 and with directed induced-fit sampling for ReplicaDock 2.0 for benchmark 5.0 targets.**
(PDF)

**S17 Fig. Compute time comparison between ReplicaDock2.0 and RosettaDock4.0.** Scaling of docking simulations on XSEDE's Rockfish Cluster for protein docking targets from the DB5.5 with respect to the number of residues.
(PDF)

## Acknowledgments

Computational resources were provided by the Extreme Science and Engineering Discovery Environment (XSEDE) and Advanced Research Computing at Hopkins (ARCH).

## Author Contributions

**Conceptualization:** Ameya Harmalkar, Sai Pooja Mahajan, Jeffrey J. Gray.

**Data curation:** Ameya Harmalkar.

**Formal analysis:** Ameya Harmalkar.

**Funding acquisition:** Jeffrey J. Gray.

**Investigation:** Ameya Harmalkar.

**Methodology:** Ameya Harmalkar.

**Project administration:** Ameya Harmalkar, Jeffrey J. Gray.

**Resources:** Ameya Harmalkar, Jeffrey J. Gray.

**Software:** Ameya Harmalkar.

**Supervision:** Sai Pooja Mahajan, Jeffrey J. Gray.

**Validation:** Ameya Harmalkar, Sai Pooja Mahajan.

**Visualization:** Ameya Harmalkar.

**Writing – original draft:** Ameya Harmalkar.

**Writing – review & editing:** Ameya Harmalkar, Sai Pooja Mahajan, Jeffrey J. Gray.

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
