## [Decision Letter · Decision Letter 0]

2 Feb 2022

Dear Prof. Gray,

Thank you very much for submitting your manuscript "Induced fit with replica exchange improves protein complex structure prediction" for consideration at PLOS Computational Biology.

As with all papers reviewed by the journal, your manuscript was reviewed by members of the editorial board and by several independent reviewers. In light of the reviews (below this email), we would like to invite the resubmission of a significantly-revised version that takes into account the reviewers' comments.

We cannot make any decision about publication until we have seen the revised manuscript and your response to the reviewers' comments. Your revised manuscript is also likely to be sent to reviewers for further evaluation.

Sincerely,

Anna R Panchenko

Associate Editor

PLOS Computational Biology

Nir Ben-Tal

Deputy Editor

PLOS Computational Biology

Reviewer's Responses to Questions

**Comments to the Authors:**

Reviewer #1: In this manuscript the authors introduce ReplicaDock 2.0, that continues the previous work on replica-exchange MC-based rigid body docking, implementing backbone flexibility and improving the scoring function. This approach aims to tackle the conformational changes via induced fit.

For each docking pose, the backbone movements are sampled over the putative interface, to both reduce computational time and capture realistic changes - this implies that the conformational change that occurs between binding partners is local, restricted to binding area. Whilst this is a possibility, it is not always the case since “off-site” conformational changes can be triggered by small interface changes and propagated to other sub-domains. It is then discussed that expanding the selection to 8A increased the <n5> across all targets during local docking. Later authors discuss that the induced fit can recover a high fraction of native contacts but the backbone is not observed to be closer to the bound conformations. For this analysis the structures generated in the low-resolution stages were used; it is unclear if the different criteria for expansion were also used here (during the low resolution stages) and how/if it would capture backbone conformations closer to the bound form.

The creation and optimisation of the motif updated dock score is in line with the objective of testing the perfomance of the method. As stated in figure S6 - “for each of the 11 protein-protein complexes.”; for completion, authors could include what were the complexes used for optimisation.

The accuracy of the optimised MUDS function is tested on 10 protein targets and it is shown to out-perform ClusPro, however the authors do state the high computational cost of ReplicaDock 2.0. It should also be included what is the total (CPU) time to run the prediction of one target, using backbone conformational sampling.

In the comparison with RosettaDock 4.0 it is shown that ReplicaDock 2.0 performs better, as expected, for moderately-flexible and and (even if modestly) to the ones with higher conformational changes. In the supplementary material the individual metrics for each complex are shown, however they could be sorted by difficulty, grouping the easy/intermediate/hard and complex side, this could could also reveal any inherent trends such as: how is the perfomance related to the complex size? A minor comment that could improve readability is that authors state that some of the complexes are “rigid”, indeed in BM5 these are named as “rigid-body” however, there is, even if, a very small (>1A) conformational change for all cases, it could be clarified in the text or renamed.

The application of the directed induced fit mechanism yields very interesting results and seems to point that the protocol could be further improving by adding information in the form of “interface patches” - could these be used in combination with the default protocol, adding perhaps weights to the “biased” interface?

ReplicaDock 2.0 is a valuable addition to the community, going in an alternate route from the common conformer-selection showing that an induced-fit approach can result in near-native molecular recognition in flexible targets. The unfortunate bottleneck is the computational cost associated with the protocol, the outlook on this limitation as well as potential optimisations could be briefly discussed.</n5>

Reviewer #2: uploaded as attachment

Reviewer #3: The authors tackle an essential problem of protein docking, which can provide crucial insights into subsequent protein functions and details on protein-protein interaction.

The described method, ReplicaDock 2.0, is a method that combines induced fit (IF) binding mechanisms with replica exchange, extending the existing rigid-docking approach with backbone motions. This allows for a more precise estimation of flexible backbone target positioning. Replica Dock 2.0 reports an impressive ~60% success rate on the targets with moderate flexibility.

Even though the paper describes a combination of the existing methods – temperature Replica-exchange Monte Carlo and Rosetta Backrub during low- and high-resolution steps of the algorithm, the overall protocol is novel and demonstrated an advantage in the prediction of protein complexes for targets with medium and- high backbone flexibility. Another notable contribution of the authors is the development of Motif Updated Dock Score (MUDS) that extends the existing Motif Dock (MDS) by incorporating knowledge-based backbone torsion statistics and Van der Waals interaction.

The authors conducted a thorough comparison of the ReplicaDock 2.0 with the existing docking methods, including deep learning-based RoseTTAFold and AlphaFold-Multimer, and defined its area of applicability. Benchmark datasets used, such as Docking Benchmark DB5.5 and recent CAPRI challenge rounds, are the community's golden standard. Deliverables are part of the Rosetta community tool, which facilitates sharing of the tool and its dissemination in the community.

There is a question regarding the docking benchmark dataset. Authors report results for ReplicaDock 2.0 and RosettaDock 4.0 on the same dataset – Docking Benchmark 5.0. The number of easy, medium, and difficult targets differ in Supplementary Table 1. It seems like three easy targets were moved to medium and difficult categories, but the rationale behind this is not clear.

Another question is regarding the computational time. Authors give a comparison in CPU hours with ClusPro, but not with the RosettaDock 4.0, another method extensively used in the paper as a baseline, or molecular dynamics, that should give more precise results but is exceedingly slow. In my opinion, listing estimates for CPU hours in global and local scope for those methods would help readers select an optimal tool for the specified level of complexity.

Lastly, I would have liked to see the authors providing a better rationale of why the approach is innovative and provides substantial (not gradual) improvement to warrant a publication in PLOS CB. If I were to play devil's advocate, it is just another docking method with no drastic improvement over other methods and is not really using a radically new method. So what warrants it to be published here?

**Have the authors made all data and (if applicable) computational code underlying the findings in their manuscript fully available?**

Reviewer #1: Yes

Reviewer #2: Yes

Reviewer #3: Yes

PLOS authors have the option to publish the peer review history of their article (what does this mean?). If published, this will include your full peer review and any attached files.

Reviewer #1: **Yes: **Rodrigo Vargas Honorato

Reviewer #2: No

Reviewer #3: No
---

## [Editor Report · Decision Letter 1]

20 Apr 2022

Dear Prof. Gray,

We are pleased to inform you that your manuscript 'Induced fit with replica exchange improves protein complex structure prediction' has been provisionally accepted for publication in PLOS Computational Biology.

Best regards,

Anna R Panchenko

Associate Editor

PLOS Computational Biology

Nir Ben-Tal

Deputy Editor

PLOS Computational Biology

---

## [Editor Report · Acceptance letter]

23 May 2022

PCOMPBIOL-D-21-02291R1 

Induced fit with replica exchange improves protein complex structure prediction

Dear Dr Gray,

I am pleased to inform you that your manuscript has been formally accepted for publication in PLOS Computational Biology. Your manuscript is now with our production department and you will be notified of the publication date in due course.

With kind regards,

Andrea Szabo
